# Developing architecture of system management in the English NHS: evidence from a qualitative study of three Integrated Care Systems

Marie Sanderson ,[1] Pauline Allen,[1] Dorota Osipovic,[1] Christina Petsoulas,[1] Olga Boiko,[2] Colin Lorne[3]

[1]Department of Health Services Research and Policy, London School of Hygiene and Tropical Medicine, London, UK
[2]Department of Health Service and Population Research, King's College London, London, UK
[3]Faculty of Arts and Social Sciences, The Open University, Milton Keynes, UK

**Correspondence to**
Dr Marie Sanderson;
marie.sanderson@lshtm.ac.uk

## ABSTRACT

**Objective** Integrated Care Systems (ICSs) mark a change in the English National Health Service to more collaborative interorganisational working. We explored how effective the ICS form of collaboration is in achieving its goals by investigating how ICSs were developing, how system partners were balancing organisational and system responsibilities, how partners could be held to account and how local priorities were being reconciled with ICS priorities.

**Design** We carried out detailed case studies in three ICSs, each consisting of a system and its partners, using interviews, documentary analysis and meeting observations.

**Setting/participants** We conducted 64 in-depth, semistructured interviews with director-level representatives of ICS partners and observed eight meetings (three in case study 1, three in case study 2 and two in case study 3).

**Results** Collaborative working was welcomed by system members. The agreement of local governance arrangements was ongoing and challenging. System members found it difficult to balance system and individual responsibilities, with concerns that system priorities could run counter to organisational interests. Conflicts of interest were seen as inherent, but the benefits of collaborative decision-making were perceived to outweigh risks. There were multiple examples of work being carried out across systems and 'places' to share resources, change resource allocation and improve partnership working. Some interviewees reported reticence addressing difficult issues collaboratively, and that organisations' statutory accountabilities were allowing a 'retreat' from the confrontation of difficult issues facing systems, such as agreeing action to achieve financial sustainability.

**Conclusions** There remain significant challenges regarding agreeing governance, accountability and decision-making arrangements which are particularly important due to the recent Health and Care Act 2022 which gave ICSs allocative functions for the majority of health resources for local populations. An arbiter who is independent of the ICS may be required to resolve disputes, along with increased support for shaping governance arrangements.

## STRENGTHS AND LIMITATIONS OF THIS STUDY

⇒ This is a qualitative study of the development of Integrated Care Systems in the English National Health Service between 2019 and 2021.
⇒ The three in-depth case studies of Integrated Care Systems include 64 in-depth, semistructured interviews, observation of eight system-level meetings and documentary analysis.
⇒ The case studies may not be representative of all national developments.
⇒ Phase 1 of the fieldwork was cut short due to the COVID-19 pandemic, which may have reduced the nuance of findings.

## POLICY BACKGROUND

Integrated Care Systems (ICSs) are a policy initiative in the English National Health Service (NHS, hereafter) whereby local 'systems' of providers and commissioners of NHS services, together with local authorities and other local partners (such as voluntary and community sector organisations), collectively plan health and care services for local populations. The approach is expected to achieve improved outcomes in population health and healthcare, reductions in inequalities in outcomes, experience and access, and enhanced productivity and value for money, in addition to helping the NHS to support wider social and economic development.[1] In stark contrast with the growing salience of ICSs, there is a paucity of empirical research concerning collaborative decision-making in ICSs in practice. It is particularly important to examine the ICS model now given the recent Health and Care Act (HCA 2022) which put ICSs on a statutory footing from July 2022, and gave them allocative functions for the majority of health resources for local populations. This paper reports a recent study examining how ICSs were developing in the

period prior to HCA 2022 and how effective the ICS form of collaboration is as a means to achieve its goals.

In order to understand the ICS model, it is necessary to first clarify ICS policy and situate ICSs within the wider context of the NHS. Alongside the use of market mechanisms to promote competition in the NHS since the 1990s, there has been an ongoing reliance on collaboration, with a long history of the development of collaborative approaches to jointly plan and deliver health, social care and public health services alongside other services.[2] Collaboration has always been an important behaviour in the English NHS, as illustrated by many empirical studies which describe the persistence of collaborative behaviour among commissioners and providers of NHS services since the establishment of the internal market.[3–6] However, while cooperation was always a feature of NHS policy and legislation, the development of ICSs has accompanied a fundamental shift away from the architecture of the internal NHS market to foreground collaboration as the dominant mode of coordination. NHS policy now describes competition as 'transactional bureaucracy' standing in the way of 'sensible decision-making',[7] and the recent legislative changes have formally removed competition as a coordinating force in the NHS.

With publication of *The Five Year Forward View*,[8] which laid out a vision to improve care delivery through breaking down barriers between different organisations and care sectors, 'integration' became a formal policy objective . This led to policy initiatives which focused on improving the coordination of service provision across organisational boundaries such as the Vanguard New Care Models Programme and the Integrated Care and Support Pioneers exemplars.[9–11] Alongside these developments, Sustainability and Transformation Plans were first introduced in 2015 as NHS organisations and local authorities (which are responsible for social care provision) were asked to work together to develop services for their local population.[12] Sustainability and Transformation Partnerships (STPs) and ICSs (a more 'mature' form of STPs) were introduced from 2016 as 'bottom-up' partnership arrangements, bringing together local organisations to deliver the 'triple integration' of primary and specialist care, physical and mental health services, and health with social care.[13] STPs were in existence until April 2021 when the last remaining STPs in England gained ICS status. For reasons of clarity, this paper will use the term ICS only.

The core tenet underlying ICSs is that the health and care needs of local populations will be best met if organisations planning and providing health and care services to that population agree collective strategies for resource utilisation. The 42 ICSs across England follow a three-tier geographically defined model (systems, places and neighbourhoods) in which collaboration at each scale addresses different aims. At 'system' scale (population size of 1–3 million covering the whole ICS footprint), collective decision-making focuses on strategic change, the development of governance and accountability arrangements, the management of performance

and collective resources, and identification and sharing of best practice. 'Places' within systems (population size of 250 000–500 000 and organised typically at borough/local authority level) are expected to focus on service integration, the development of anticipatory care, out-of-hospital care and hospital discharge. 'Neighbourhoods' (population size of 35 000–50 000 and based around non-statutory Primary Care Networks of groups of general practitioner (GP) practices) are expected to improve integration of primary health services with community healthcare services and other local health and care organisations. In practice, systems (and 'places' and 'neighbourhoods') vary considerably in terms of population size and organisational complexity, reflecting local factors such as demography and existing networks of collaboration, and may elude neat containment within coherent territorial geographies.[14]

It is particularly important to examine how ICSs are developing as the 'system' has become the central mechanism through which the achievement of NHS goals is coordinated. Systems are expected to develop coordinated plans for NHS activity, workforce and money. The approach taken by the NHS economic and structural regulator—NHS England and Improvement (NHSEI)—is tailored to give primacy to the system in financial and performance matters, alongside NHS organisations' individual accountabilities (which remain unaffected).[15] Additionally, financial rewards are being linked to system rather than individual organisation performance, such as linking the attainment of system financial targets to financial rewards for individual NHS organisations.[16]

ICSs have recently become even more significant bodies. The recent HCA 2022 put ICSs on a statutory footing from July 2022, consisting of a dual structure of a statutory body, the Integrated Care Board (ICB) (focused on integration within the NHS and accountable for NHS resources), and a statutory committee, the Integrated Care Partnership (focused on integration between NHS, local government and wider partners). Clinical Commissioning Groups (CCGs) (formerly the commissioning bodies) were abolished with the transfer of allocative functions to the ICBs. Consequently ICBs now have responsibility for commissioning acute, community and mental health NHS services for their population, primary medical care, and possible further delegations from NHSEI including other primary care budgets.

It is important to understand collaboration within the wider institutional context. Of particular importance in relation to ICS policy is the permissive nature of governance arrangements. ICSs have considerable freedom to decide their own local governance arrangements rather than following a prescribed national blueprint. At the time of the research, each ICS could tailor governance arrangements to suit local circumstances, within minimum governance requirements for a 'Partnership Board' which provides a forum for collective action on issues that affect all system members,[13] and this minimal and permissive approach remains the case under the

HCA 2022. The permissive nature of local governance has significant implications when coupled with the principle of subsidiarity (where decisions are taken closest to those affected). This is particularly so in light of HCA 2022 which carries the expectation that statutory ICBs will delegate substantial decision-making regarding the allocation of resources to committees and subcommittees, such as 'place-based committees' and provider collaboratives (non-statutory partnership arrangements involving two or more trusts),[17 18] for which there are no national governance requirements. It is therefore important to understand how ICSs are addressing the challenge of agreeing local governance arrangements while addressing the principle of subsidiarity.

A second important aspect of ICS collaboration relates to organisational sovereignty. Collaboration necessarily remains a voluntary, consensual, non-binding model of coordination (although effectively mandated by NHS policy for NHS organisations), and providers remain separate organisations with their own organisational interests and accountabilities, and freedom to dissent. All system partners have their own accountabilities and statutory responsibilities which they must hold in regard when agreeing collective system plans. For example, NHS Trusts and Foundation Trusts (FTs) have legal duties to provide safe care and treatment (HSCA 2008) and FT boards have a duty to act with a view to promoting the success of the Trust to maximise the benefits for the members of the Trust as a whole and for the public (HSCA 2012). NHS Trusts and FTs have direct accountability to NHS England for their performance. System partners from outside the NHS, such as local government or independent sector organisations, are subject to separate institutional contexts regarding priorities, ways of working and financial rules.

Third, ICSs exist in a complex landscape of pre-existing partnerships and planning networks which must be accounted for, such as Health and Wellbeing Boards (formal committees of local authorities, which have a statutory duty, with CCGs, to produce joint strategic needs assessments and joint health and well-being strategies for their local population).

These complexities raise questions about how collaborative decision-making in ICSs will work in practice, including the extent to which organisational sovereignty disrupts the ability of systems to achieve a consensus. Now that the HCA 2022 has come into force, ICSs have significant allocative responsibilities, and are subject to associated expectations including of improved outcomes.[1] To make headway with this agenda, ICSs will need to agree with suitable local governance arrangements to discharge their functions according to the principle of subsidiarity, and make challenging collective decisions, which may be perceived as disadvantaging individual members. It is important to examine how these issues have been experienced and addressed in ICSs to date. A small number of empirical studies have been published which are concerned with the development of collaborative

arrangements within ICSs,[19–24] and the development of commissioning in the light of system collaboration.[25 26] The study reported in this paper makes a significant contribution to this empirical evidence by providing a nuanced analysis of the development of governance, accountability and decision-making arrangements in three ICSs.

ICS policy does not explicitly draw on theory to explain how the use of collaborative decision-making processes will lead to the attainment of ICS aims such as enhancing productivity and value for money. Perspectives from political science and public administration can be deployed to analyse the development of collective action in ICSs, or to facilitate successful collective action, such as Jones et al's use of Ansell and Gash's conceptual model of collaborative governance to inform the development of the role, behaviour and skills of medical leaders of ICSs.[27 28] We have chosen to focus on the work of Ostrom,[29 30] rooted in economic theories of cooperation, which suggests that, contrary to the received wisdom of 'the tragedy of the commons', communities can cooperate to self-manage limited shared ('common pool') resources in a way that benefits all community members and leads to the sustainability of the resource. Ostrom's conceptualisation of common pools as limited natural or man-made resource systems on which multiple parties depend has resonance with collectivism and universality of public services in the context of finite resources.[31 32] The development and functioning of system working in the English NHS in which local 'systems' are required to adopt collective resource utilisation strategies to manage a finite local pot has evoked connections with the work of Ostrom, and led to the use of her theories as an analytical framework to understand the development of system working.[33 34]

A cornerstone of Ostrom's work is her design principles which describe the conditions required for communities' successful self-governance of common pool resources. The principles address the need for 'communities' to set up clear boundaries and membership, agree for themselves rules regarding how resources will be used, establish a balance between costs and benefits of collaboration, and agree the process for monitoring of behaviour and sanctions.[29] The principles also allow that wider context, referring to the broader contextual variables in which collaboration takes place, can enable or inhibit collaboration, for example, monitoring, enforcement and sanctioning institutions, and the relationships between actors. Ostrom's design principles are of value both as a 'heuristic' to guide collective approaches to the planning and delivery public services,[35] and as an analytical frame through which to interpret collective approaches. This paper draws on these design principles as a frame to help understand the ways in which ICSs and the wider context in which they are situated support the development of collaborative decision-making through the system approach.

## Study questions

Our research questions were based on our understanding of ICS policy, and the literature regarding economic theories of cooperation, in particular the work of Ostrom.[30] The questions focus on three broad areas: first, how decisions are being made in ICSs; second, how ICS partners are balancing collective and individual interests; and third, what kind of decisions systems are making regarding the allocation of resources.

In relation to the first area, how decisions are being made in ICSs, we wanted to establish: how the local leadership and cooperative arrangements with stakeholders (statutory, independent and community-based, including local authorities) were governed in light of policy recommendations. Second, in terms of the balancing of collective and individual interests, the study addressed: how individual organisations are reconciling their role in an ICS with their individual roles, accountabilities and statutory responsibilities. Third, we wanted to establish: what decisions regarding the allocation of resources are being made through ICSs, in particular whether ICSs are able to allocate resources more efficiently across sectoral boundaries and bring their local health economies into financial balance.

Our research was divided into two phases. The first phase focused on the system scale. In the second phase of our research, we addressed similar questions while focusing on the development of 'place-based partnerships', and the developing role of the regional NHSEI function (regional teams which are responsible for the quality, financial and operational performance of all NHS organisations in their area).

## Study design

The study used qualitative methods with an additional quantitative component. The results of the quantitative analysis are included in our final report.[36] Primarily, we used a case study research design, consisting of three in-depth case studies, each consisting of a system and its partners. The use of case studies was thought to be the most appropriate research design for this study as interviews and documentary analysis were informed by the contextual information we were able to gather by concentrating on three specific systems. An initial literature review of NHS system governance[37] was drawn on to inform strategy when selecting case study sites. This literature review led to the identification of various characteristics of interest in local contexts which might be important in relation to how system working developed. These included: the number and variety of providers of NHS services in the system; the number of local authorities within systems; and the degree of fit between health and local authority boundaries. We shortlisted systems which had one or more of the following characteristics: system boundaries which did not correspond to local authority boundaries; the presence of private sector and/or social enterprise partners; a concentration of providers; a concentration of local authorities. From our shortlist, we sought to recruit

**Table 1** Phase 1 interviews by case study site and organisational type

| Organisation | Case study 1 | Case study 2 | Case study 3 | Total interviews |
|---|---|---|---|---|
| ICS leadership | 2 | 4 | 2 | 8 |
| CCG | 0 | 1 | 1 | 2 |
| NHS providers | 3 | 3 | 4 | 10 |
| Local authorities | 1 | 1 | 4 | 6 |
| Primary care | 0 | 0 | 0 | 0 |
| Other providers | 0 | 2 | 0 | 2 |
| Total interviews | 6 | 11 | 11 | 28 |

CCG, Clinical Commissioning Group; ICS, Integrated Care System; NHS, National Health Service.

case study sites which demonstrated variance across these characteristics. Additionally, as we were also interested in the role of the regional NHSEI function, we sought to select case study sites from differing NHSEI regions. In phase 2 of the research, a single 'place' within our three case studies was identified based on characteristics of interest emerging from the phase 1.

The first phase of fieldwork was undertaken between December 2019 and March 2020 and focused on studying ICSs (and their predecessor STPs). Fieldwork was interrupted in March 2020 by the COVID-19 pandemic. In particular, we had fewer interviewees in case study 1 (CS1) than intended. The second phase of fieldwork took place between January 2021 and September 2021 and focused on a more detailed examination of a selected 'place' within each of our case studies. All interviews in the second phase of the fieldwork were conducted over an online platform rather than face to face. We conducted a total of 64 in-depth, semistructured interviews (see tables 1 and 2) and observed eight system-level meetings (three in CS1, three in case study 2 (CS2) and two in

**Table 2** Phase 2 interviewees by case study site and organisational type

| Organisation | Case study 1 | Case study 2 | Case study 3 | Total interviewees |
|---|---|---|---|---|
| ICS leadership* | 2 | 2 | 3 | 7 |
| Regional NHSEI | 1 | 1 | 1 | 3 |
| CCG | 3 | 0 | 5 | 8 |
| NHS providers | 2 | 2 | 3 | 7 |
| Local government | 1 | 2 | 3 | 6 |
| Primary care | 1 | 1 | 1 | 3 |
| Other providers | 0 | 1 | 0 | 1 |
| Other | 0 | 1 | 0 | 1 |
| Total interviews | 10 | 10 | 16 | 36 |

*Where an interviewee held a joint ICS/CCG role, this is recorded as an ICS leadership interviewee.
CCG, Clinical Commissioning Group; ICS, Integrated Care System; NHS, National Health Service; NHSEI, NHS England and Improvement.

case study 3 (CS3)). Interviewees were recruited due to their role as senior management representatives of system partners who participated in the main decision-making forums at system scale, and within the selected 'place'. All participants gave informed consent. Topic guides related to the study questions described above. The purpose of observing a variety of meetings was to supplement the information we obtained from interviews. In addition, we gathered documentation from all three case study sites which included strategic plans, meeting papers and details of governance structures. These sources were used to add detail to the interview accounts.

The three case study sites (which consisted of one ICS and two STPs at the time of recruitment) are located in different parts of England. CS1 covers an urban population, has complicated boundaries and includes five unitary authorities. It gained ICS status in 2021. CS2 system shares near coterminosity with the county council, and system partners include social enterprises. It gained ICS status in one of the earliest waves. CS3 system has a large geographical footprint, and a complex, multilayered governance structure spanning seven CCGs (merging to a single CCG in 2021) and eight local authorities. It became an ICS in 2020. The change in status from STP to ICS in CS1 and CS3 during the fieldwork did not impact our data collection as system members and leaders, and the ongoing work of the system remained unaltered.

PA, MS, DO and CL agreed the theoretical framework, and the main themes derived from the research questions. MS, DO and CP agreed additional themes emerging from the data. The initial themes for our analysis included: partners' definition of the system and membership; the structure of governance arrangements; perceptions of developing accountabilities; developing spatial scales and functions; system resource allocation; relationships; drivers of cooperation; use of competition; devolution and space to act. The analysis of phase 2 data drew on the same themes, with the addition of a theme concerned with the future development of system working. The interviews were transcribed and coded (by MS, DO, OB, CL and CP) using the agreed coding framework. The principal researchers (MS, DO and CP) met periodically to check whether the coding framework was working well, to discuss emerging findings, and check researchers' interpretation of the data and areas of difference between the case studies and to agree to any necessary modifications to the coding framework.

### Patient and public involvement

No patients or members of the public were involved in this study.

## RESULTS

Our findings are grouped into three sections, each relating to a significant aspect of ICS decision-making. First, the development of decision-making arrangements in ICSs; second, how organisations are reconciling systems and individual roles; and third, the kind of decisions ICSs are making regarding the allocation of resources.

### Development of decision-making arrangements

System partners were generally enthusiastic about the value of increased collaboration, seeing this as the best way to achieve better use of resources and health improvement across health and social care. The views of local authorities were mixed, viewing system development as both an opportunity and with a dose of scepticism. They were keen to be involved in arrangements as an equal partner, and not the 'last thing that you come to' in a health-focused system (local authority director 4, CS3). Other non-NHS partners (social enterprises in CS2) also viewed ICSs with scepticism, for example, the emphasis on achieving financial balance in the NHS was seen by some as illustrating the NHS-centric focus.

The refinement of governance arrangements was an ongoing task for local partners. Part of this task was agreeing the spatial configurations of systems and 'places'. We found that agreement between health and local government of the 'best' spatial configurations was of particular importance to ensuring clarity of governance arrangements. In two of our case studies (CS1 and CS2), local partners appeared to be in agreement regarding the most sensible system and 'place' configurations. In CS3, however, where the system spanned seven CCGs (merging to a single CCG in 2021) and eight local authorities, trying to reach a consensus among partners about 'place' configuration was a lengthy process, making it difficult to progress integration, a process described as 'building the aeroplane while flying it at multiple levels' (NHS Trust director, borough-based partnership 1, CS3). In CS3, local government configurations were perceived to be a particularly awkward fit at the system level due to the sheer volume of organisations involved. Local actors deviated from the system/place division in favour of a 'double-layer' set-up, exemplified by the presence of an intermediate subsystem level which lay between the lower-tier place partnerships (corresponding with local authority boundaries) and the ICS, described by one interviewee as 'systems within systems within systems' (local authority director 1, CS3). This arrangement was thought to reflect more accurately local configurations, but was also acknowledged, due in part to the lack of uniformity, to remain complex, risking confusion and lack of clarity. In this case study, the role and membership of governance forums were differently understood and described, and the future shape of governance arrangements was contested.

Beyond the local agreement of spatial configurations, system partners were finding agreeing local governance arrangements inherently challenging. This was seen to reflect both the scale of the system agenda and the already complex institutional landscape in which ICSs were situated:

Achieving clarity over where you make decisions, who makes decisions, and then who enacts them is really difficult, and you often only find out you've got it wrong by doing it…this is bottom up, and it's to take into account statutory body decision making, trying to make use of architecture that was already there, and then linking it all together. And every time we do it, we find other bits that we then add in, because it's just reflective of the size of the remit of an ICS. (ICS director 1, CS2)

The drive to establish partnership working at the lowest possible level, in line with the principle of subsidiarity, was hampered by a lack of clarity both from national policy and locally on how to distribute power, resources and responsibilities between different levels of governance. Local actors in all three case studies found it challenging to decide which decisions and functions should sit where. In particular, in CS3, the agreement of such arrangements was further hampered by the lack of consensus regarding the configuration of 'places', reflecting the existence of two non-aligned spatial configurations at 'place scale'. In all the case studies, going through these arrangements locally on a case-by-case basis was a time-consuming and complex process, which was particularly difficult given the shifting sands of policy, the prioritisation of the COVID-19 response and, in some instances, the existence of power dynamics regarding who the decision-makers were.

Increasingly, formal governance arrangements were being developed which included an emerging focus on the prioritisation of 'place' collective voice over representation of individual organisations. All of our case studies were considering the adoption of a formal partnership arrangements in 'places', such as an alliance agreement, although only one (CS2) had adopted a formal alliance agreement. There was some frustration regarding the effort expended on the establishment and refinement of governance and the perceived added value of this activity. As the lead of a place-based partnership observed, informal relationships between partners were more important to the achievement of collaboration than formal governance arrangements:

I think you can easily really get quite led astray on the governance. You can easily spend years and years doing the governance. But I think in reality it's very difficult in governance terms and in NHS contracting terms to force an organisation to do something they don't want to do, and actually in all my years, and I've got many years, actually, in reality I've hardly ever voted on a board, hardly ever had to have a count up of those, and I've hardly ever gone through any sort of legal proceedings on NHS contracts. (place director, CS2)

Others experienced governance architecture as significant. For example, smaller partners such as GPs, and those who were not often previously invited to the table,

such as District Councils, welcomed the formal structures which allowed them an equal voice in discussions.

### Reconciliation of system and individual responsibilities

The reconciliation of system and individual responsibilities was reported similarly across the three case studies. This reconciliation was aided by an ongoing shift from competition to collaborative working, and a changing environment regarding commissioning mechanisms, pricing structures and financial incentives. In the second phase of the research, the changing financial regime in response to COVID-19 was reported to have 'completely rewritten the rulebook' (ICS director 2, CS2), moving to block contract payments 'on account' for all NHS providers, with suspension of the Payment By Results (PBR) national tariff (PBR is a prospective payment system, associated with incentives for competition, in which each episode of care is charged at national tariff rates). In all case studies, formal tendering or competitive processes were no longer anticipated to be a commonly used commissioning mechanism.

While incentives for competition among providers had subsided, organisations were still finding it challenging to balance system and individual responsibilities. Among NHS partners, there was scepticism about the effectiveness of financial incentives to encourage NHS organisations to favour a system perspective. In the first phase of our research, the notion of achieving financial balance within systems was widely viewed as unrealistic, unattainable and unsupported by the wider regulatory context. More detailed objections were that individual control total allocations did not consider local circumstances and imposed stringent efficiency targets on already struggling and historically underfunded providers. Agreeing projections of performance against control totals was described as a process of negotiation with NHSEI. In the second phase, interviewees were concerned that while the Elective Recovery Fund (additional funding for clearing the elective backlog created by COVID-19) was encouraging organisations to make plans together, it was not a sufficient mechanism to stop individual organisations giving priority to their organisational interests and patients. One acute trust director saw a clear tension between 'the glib [regional NHSEI] vision that we've all suddenly switched to managing waiting lists as a sector' and what they saw as the duty of NHS Trusts to prioritise their own patients:

There's a huge variation in the scale and nature of the problem in the different organisations, and we at [hospital] hold most of the problem on elective recovery in terms of the long waits. And if everybody were to suddenly use all their capacity then, for the good of the system, some organisations wouldn't do any operating on their own patients for a very long time, they would spend a long time operating on our patients and not much else. And that's not really a proposition that you can put to the statutory body and expect it to accept that, so while we're making

incremental steps in that direction, they know that's not feasible. (director, acute trust, CS3)

Provider concerns that system priorities could run counter to organisational interests were prevalent. On the one hand, some interviewees were quite sanguine about the prospect of dropping some of their organisational priorities in favour of shared priorities, if this led to an improvement of services in the locality. For example, an acute trust director suggested that the trust would be prepared to spend extra money on areas of need, such as housing, and other services rather than spending it on their own hospital. Others, however, reflected on the potential risks of collective decision-making in the light of individual organisation's statutory responsibility to ensure that risks to the organisation and the public were mitigated effectively. One acute trust CEO summarised it:

So then you get into a conversation, well, maybe there's horse trading to be done in the system, which is I expect what the centre thinks, they think, well, they will just have to agree across the system to cut their cloth if you like…X Hospital needs a new roof which is more important than my theatres because the rain gets in on the patients…I mean, if a woman in my organisation dies of some hideous infection after she's had her section, I wonder who's going to be in the coroner's court explaining why we let her be operated on in an operating theatre that I knew wasn't meeting the standard. It's really tricky, isn't it? (director, acute NHS FT, CS2)

A further perspective on balancing system and individual priorities was provided by local authority and the independent sector interviewees in CS2. From the local authority perspective, the wider institutional context was not conducive to system working due to differences in business and planning cycles between health and local government, the wider remit of local councils (of which social care was only a part) and differing approaches to procurement. Where system or 'place' boundaries were not aligned with local authority footprints such as in two-tier 'place' configuration in CS3, local authorities were more reluctant to engage in strategic commissioning and planning discussions. Local authority interviewees in all case studies were also concerned about their potential exposure to financial risk, and loss of control over limited council resources. Interviewees from the two social enterprises in CS2 suggested that balancing individual and system roles was very difficult for independent sector organisations, who had obligations to break even and sat outside the supportive policy context of the NHS.

System partners in all case studies acknowledged that, as system commissioning responsibilities evolved, conflicts of interest were inherent in this partnership mode of decision-making, but believed that the benefits of collaborative decision-making outweighed the risks of conflicting interests. In terms of overcoming conflicts of interest, it was thought that conventional methods of addressing conflicts, most commonly by removing the conflicted party from the decision-making process, were insufficient as everyone was an interested party with a potential conflict. It was hoped that the close collaborative environment and peer monitoring would guard against abuses of influence, and that the consensus model of decision-making would allow objections to be voiced.

Accountability is a central concept when examining the potential of ICSs to achieve their goals, both vertical (and formal) accountability (holding to account of the system, system leaders and (NHS) system partners for system performance by NHSEI), but also informal and horizontal accountability (the holding to account of system partners by the system). ICSs also have an informal accountability relationship with the public which should be considered alongside system partners' own accountabilities to the public. Horizontal accountability between system partners was reported across our case studies to be weak, characterised by 'softer' mechanisms of holding to account through trust, rather than in a formal or codified way. This developing assurance function concerned open information exchange about organisational performance, quality and finance which could facilitate open discussion and serve as an incentive to improve.

An understanding of the needs of local patients and communities underlies the aims of systems, particularly those around population health and the development of local partnerships. The case study systems were developing routes to public engagement of various kinds and at varying spatial scales, seeking to understand the priorities, needs and preferences of the population. Each had established citizens' panels with varied aims, such as in CS1 to start a public debate about allocation of limited resources. Other routes to engagement included research to understand residents' opinions and activities in conjunction with Healthwatch. At the time of the fieldwork, ICSs had no formal accountability to the public. Formal accountability was understood to lie with, and largely be performed through, the partners that held a legal duty to involve the public. It was acknowledged this meant the visibility to the public of the ongoing work of the collaborative partnerships and hence public accountability remained low.

### Decisions regarding resource allocation being made by systems

Our research was conducted during the early days of system working, and due to the disruption caused by the COVID-19 pandemic, it is difficult to assess the extent to which ICSs are achieving their aims concerning the allocation of resources more efficiently and financial balance within the system. We gathered multiple examples of work being carried out across systems and 'places' to share resources, change resource allocation and improve partnership working (see table 3 below for examples of work at place scale). However, local actors acknowledged that the impact of these initiatives in terms of efficiencies and quality markers is difficult to quantify.

**Table 3** Examples of work being carried out at place scale

| Case study | Examples of partnership working in 'places' |
| --- | --- |
| CS1 | Development of data-driven approach to care<br>▶ Establishment of population health unit across local authority and acute trust<br>▶ Data sharing across primary and secondary care<br>Appointment of health ageing coordinators across social, primary and secondary care<br>Development of system-wide pathways, such as end-of-life care strategy |
| CS2 | Resolution of operational performance issues, including day-to-day capacity management<br>Work with wider partners to situate services outside hospital, including development of new premises<br>Development of key worker affordable housing on hospital site<br>Development of opportunities for shared service delivery, such as urgent treatment centre<br>Decisions regarding the distribution of non-recurrent funding<br>Development of 'integrated delivery units' such as discharge team with jointly funded lead<br>Pilot for 'step-down' nursing provision to aid hospital discharge |
| CS3 | At intermediate subsystem tier:<br>▶ Sharing best practice across boroughs<br>▶ Performance management and assurance<br>▶ Resource allocation<br>▶ Operational command for COVID-19<br>In borough-based partnerships:<br>▶ Development of 'multidisciplinary discharge hubs'<br>▶ Pathway development for interface between hospital and wider system<br>▶ Operational collaboration during COVID-19 response<br>▶ Development of shared workforce strategy<br>▶ Decisions regarding the distribution of COVID-19 contingency funding |

CS1, case study 1; CS2, case study 2; CS3, case study 3.

At system scale, agreements had been reached to share resources in order to take advantage of economies of scale, and offer mutual support. A common focus was sharing staff (both managerial and clinical) between providers with a view to helping to improve performance, sharing best practice and expertise, joint staff bank and a virtual academy. CS2 appeared most proactive in sharing resources at system and place level, and this had in part been enabled by considerable transformation monies associated with early ICS status which had been used to pilot changes to care design and delivery. In all case studies, further sharing of resources was necessitated by the pandemic, where partners made collective decisions about allocating funds and risk-sharing in the course of the pandemic response. It was recognised, however, that the real test about sharing of resources would come in the future, when decisions about priorities would need to be taken in normal conditions rather than either in the middle of a pandemic or accompanied by significant additional funds.

As described in the section above, the financial regime changed greatly during the period of the research, moving towards the facilitation of collaborative behaviour. While these changes in payment mechanisms were seen as helpful facilitators, collaboration around the collective use of resources was not plain sailing. Other forms of competition between providers remained, for example, competition for allocation of resources or competitive

pressures in distribution of services, access to workforce, capital and investment.

Overall, the changing financial regime did not appear sufficient to allow systems to address long-standing issues. While systems were engaged in negotiating actions to achieve long-term financial sustainability, for example, to spend more in primary/community services, increase digital interventions, reduce duplication of functions across organisations and limit ineffective procedures, this had not yet translated into specific agreements in practice. In CS2, forthcoming work to decide functions to be shared across acute hospitals, and reduce face-to-face outpatient appointments, was expected to be a 'really difficult and painful' process (ICS director 3, CS2).

Some interviewees reported there was reticence addressing such difficult issues, such as the need to reconfigure services across sites to make savings, in ICS forums due to the decision-making model. The CS2 ICS accountable officer suggested organisations' statutory accountabilities were allowing a 'retreat' from the confrontation of difficult issues facing systems, such as agreeing action to achieve financial sustainability. Place-based partnerships, due to the informal nature of their working, were not seen as an appropriate forum for disagreement and difficult discussions. An acute trust director in CS2 noted it was difficult to discuss performance issues in 'place', such as a reported lack of GP appointment availability causing an increase in demand for urgent care in hospital, particularly at a time when service providers were under a great deal of strain due to the response to COVID-19, and in light of voluntary nature of cooperation.

## DISCUSSION AND CONCLUSION

Our findings suggest that the shift to collaborative working has been largely welcomed. While this was particularly the case for NHS organisations, other system partners, specifically local authorities and non-NHS providers, welcomed the shift to collaboration, but were more critical of the vehicle of ICSs due to the perceived NHS-centric focus of ICS policy.

Wider context, referring to the broader contextual variables in which collaboration takes place, can enable or inhibit collaboration.[38] The institutional context in the NHS is reshaping to accommodate collaborative approaches: commissioning mechanisms, pricing structures and financial incentives are subject to change, along with regulatory approaches. While progress in achieving system aims had been hampered by the operational response to the COVID-19 pandemic, local actors felt that collaboration in systems led to improvements in ways that did not occur previously and, in particular, cited many examples of changes to service delivery that had been achieved through place-based partnerships. However, our findings suggest there are challenges in making decisions through ICSs, particularly in relation to reaching agreement concerning complex and/or difficult matters. These challenges need to be recognised as statutory ICBs

enact their allocative responsibilities, and the complexity and scale of ICS activities and decisions increase.

This study, based on case studies of three ICSs, provides a detailed and nuanced analysis of the ongoing development of ICSs, and the effectiveness of this form of collaboration as a means to achieving ICS goals. This is particularly important and timely given the recent legislation changes of HCA 2022 from July 2022. The study has certain limitations. First, phase 1 of the fieldwork (conducted between December 2019 and March 2020) was cut short due to the COVID-19 pandemic. We were not able to interview all partners in our case studies. In particular, we had fewer interviews in CS1 than intended. This restriction may have reduced nuance in the findings of this report. Second, as the study design consisted of three in-depth case studies, it is not possible to make statistically based generalisations to the whole NHS. However, as the study is based on a strong theoretical framework, it is possible to make analytical generalisations. We have noted the extent to which findings from the three case studies themselves converged and diverged. Third, given the disruption of the pandemic, it is very difficult at this time to evaluate the extent to which ICSs are going to be able to allocate resources more efficiently across sectoral boundaries and bring their local health economies into financial balance.

Our findings suggest there remain significant challenges regarding agreeing governance, accountability and decision-making arrangements which need to be addressed to facilitate successful collaboration. Factors identified by Ostrom as necessary building blocks for successful collaboration, such as agreeing clear boundaries and membership and agreeing how decisions should be made, were proving difficult to address in some systems. Earlier studies of systems[19 24] found attention in developing STPs and ICSs was focused on ground work and preliminary activities, and it is notable that system governance arrangements are still subject to ongoing refinement. Our research suggests where complexity in the local context increases, particularly where there is a no 'natural fit' between the health and local government footprints, it can be very difficult for partners to agree governance arrangements. This is a particular risk in relation to partners outside the NHS, most pertinently local government, where there is weaker incentivisation in the first place to engage with system working. Where system and local council footprints aligned (as in CS2), statutory planning bodies involving local authorities, such as Health and Wellbeing Boards, could become incorporated into system architecture. CS3 was distinct as an illustration of the difficulties encountered where system and place spatial scales are not considered as coherent or meaningful groupings across health and local government. Our findings suggest that awkward boundaries can threaten local government 'buy-in' to strategic commissioning and planning discussions. Negotiations among multiple parties to achieve clarity about governance arrangements drain resources and consume time. Furthermore, where

governance arrangements are not considered coherent or meaningful, this can limit engagement of partners.

There is a balance to be struck between retaining flexibility at ICS level regarding governance arrangements, and having to follow national guidance. It has been noted that the ambition for local flexibility in HCA 2022 is encouraging as it is considered a key enabler of collaboration, and there are hopes this flexibility will be protected from 'the NHS's tendency to centralise, which could lead to an overly prescriptive system architecture—despite everyone's best intentions.'[39 40] A key tenet of Ostrom's design principles is that, for collaboration to be successful, local parties need to be involved in the development of the rules of the game.[30] The iterative development of governance arrangements among local parties is thought to be important in developing norms of trust and reciprocity between partners which underpin increased collaborative working, and encourage fairness and adherence to local rules.[30] However, where a similar process is occurring in parallel ICS, it can also be argued that 'reinventing the wheel' should be minimised. There is a case for increased support for systems in their task of putting in place clear 'rules of the game', including additional specified 'scaffolding' shaping governance requirements such as committee membership and accountability arrangements, to avoid unnecessary local discussion where local areas are all engaged in similar tasks. This is particularly pertinent in light of the lack of specification in HCA 2022 and associated guidance regarding governance arrangements in place-based partnerships or provider collaboratives where it is anticipated many ICB functions will be delegated. Local 'fatigue' regarding the ongoing refinement of governance arrangements should be acknowledged, together with the possibility that this fatigue may outweigh relational gains particularly where there are existing strong relationships.

Despite changes in the NHS institutional context to support adoption of 'best-for-system' perspective, the reconciliation of system and individual responsibilities is proving difficult in the light of organisational sovereignty and the lack of formal authority of system leaders. Ostrom suggests that for collaboration to succeed, participants should feel the costs and benefits of collaboration are in balance. Our findings indicate that partners are not convinced that the separate statutory obligations of individual organisations would always be best served by taking decisions on a best-for-system perspective. This echoes findings of earlier studies of ICSs and their predecessors, STPs.[24 41] Indeed, in their study of STPs, Waring et al found that, far from putting interests aside, partners were engaged in 'micro-political' disagreements seeking to advance or protect their particular preferences, agendas or interests.[41] Such disagreements indicate the challenges of addressing contentious issues in the light of organisational sovereignty without independent arbitration and hierarchical control.

Importantly, making ICSs statutory bodies does not overcome this problem, as partner organisations will retain

their organisational sovereignty, and consequently the capacity to disagree with system-proposed plans. There are a number of possible avenues to address this problem. One strategy is to develop strong horizontal accountabilities between system partners allowing them to develop the necessary sanctions to build trust and ensure adherence of agreed 'rules of the game'.[30] Our research indicates that such structures are currently underdeveloped, and it is unclear how well those new lines of accountabilities, especially the horizontal ones, will work in practice. A further potential strategy, as proposed by Waring *et al*, is, given the absence of formal authority in ICSs, to seek to improve system leaders' political skills, developing negotiation and deal-maker skills to identify means of offsetting perceived losses.[41] It is also possible that the issue may be further addressed through changes in HCA 2022 which seek to change the policy context, incentivising the adoption of a 'best-for-system' approach by introducing a 'duty to cooperate' for NHS bodies and a 'triple aim' duty to consider the effects of their decisions on the better health and well-being of everyone, the quality of care for all patients and the sustainable use of NHS resources. Given the inherently voluntary, consensus-driven nature of collaboration, it is likely that a combination of all the above approaches will be necessary to assist systems in making contentious decisions. It is also our contention that an arbiter independent of local system members may be still required to resolve disputes and it seems likely that the regional directors of NHSEI could undertake this role in practice.

Looking ahead, under HCA 2022, the collaborative approach will be applied to decisions regarding the allocation of resources. Our research raises a number of points in this regard. First, the tensions in decision-making in ICSs, particularly concerning addressing difficult issues, together with a lack of formal arrangements to deal with disagreements, could become significant fault lines as statutory ICBs enact their new commissioning responsibilities. Second, conflicts of interest in relation to commissioning decisions will be pervasive with no clear route for mitigation. Although interviewees felt negative consequences were outweighed by the benefits of collaborative decision-making, arguably this issue goes to the heart of how ICBs will be able to operate in the interests of the local population as opposed to prioritising those of powerful organisations. It is not clear how, in the absence of a separate commissioning body whose sole role it is to achieve results without having undue regard to the effects on the finances of individual organisations, ICBs will be able to plan and commission services which best meet the needs of local populations. It is not clear that using the ICS model consensus will always be achieved, nor that it will be the optimum consensus for population health.

In conclusion, while the ICS model of collaboration has been embraced by local actors in the NHS and elsewhere, there remain significant challenges regarding agreeing governance, accountability and decision-making arrangements. Viewing ICSs through a network governance or collaborative governance perspective such as that of Ostrom's work is a valuable approach to assess the development of collective action in the articular context of ICSs, and to identify measures which might be taken to strengthen arrangements. It is clearly important to continue to study the development of system working in the future to see how these issues are tackled as the effect of the pandemic diminishes and systems have longer experience of working together.

**Contributors** All of the authors (MS, PA, DO, CP, OB and CL) were involved in the design and data analysis of the study, and contributed to the drafting, revision and finalisation of this paper. In addition, OB, CL, DO, CP and MS collected the data for this study. PA is responsible for the overall content as guarantor.

**Funding** This research is funded by the National Institute for Health Research (NIHR) Policy Research Programme, conducted through the Policy Research Unit in Health and Social Care Systems and Commissioning (PR-PRU-1217-20801).

**Disclaimer** The views expressed are those of the authors and not necessarily those of the NIHR or the Department of Health and Social Care.

**Competing interests** All of the authors received grant funding from the Department of Health via its Policy Research Programme for this research. No authors have had financial relationships with any organisations that might have an interest in the submitted work in the previous 3 years, and no authors have any other relationships or activities that could appear to have influenced the submitted work.

**Patient and public involvement** Patients and/or the public were not involved in the design, or conduct, or reporting, or dissemination plans of this research.

**Patient consent for publication** Not required.

**Ethics approval** NHS research governance approval from the HRA was granted on 6 August 2019 (266175/REC ref 19/HRA/3261). Ethical approval for the study was granted by the London School of Hygiene and Tropical Medicine internal ethics committee on 23 August 2019 (ref: 17711). The research received endorsement from the Association of Directors of Adult Social Services Executive Council on 19 November 2019. Participants gave informed consent to participate in the study before taking part.

**Provenance and peer review** Not commissioned; externally peer reviewed.

**Data availability statement** No data are available.

**ORCID iD**
Marie Sanderson http://orcid.org/0000-0001-7917-6786

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
