## [Reviewer comments · BMJ Open]

ARTICLE DETAILS

TITLE (PROVISIONAL)	The developing architecture of system management in the English NHS: evidence from a qualitative study of three Integrated Care Systems
AUTHORS	Sanderson, Marie; Allen, Pauline; Osipovic, Dorota; Petsoulas, Christina; Boiko, O.; Lorne, C.

VERSION 1 – REVIEW

REVIEWER	Hughes, Gemma University of Oxford, Nuffield Department of Primary Care Health Sciences
REVIEW RETURNED	14-Sep-2022

GENERAL COMMENTS	This was a well written and very clear account of the development of Integrated Care Systems in England. The main point that needs to be addressed was the noticeable absence of patients and the public in this paper - where are their voices in ICSs? This needs to be considered, even if simply as a limitation of this paper (in terms data/lack of data in the cases and in analysis as well as PPI in the paper itself). I have some other minor comments that the authors could consider addressing, as I think these points could improve an already very interesting paper. The policy background is very clearly set out, however I would like a little more history included (or relevant citations) about the change from competition to collaboration – are ICS really the marker towards collaboration the authors suggest? I think there is a longer history of trying to balance collaboration and competition that dates back at least to the 2012 Health and Social Care Act, see for example the Nuffield Trust briefing ‘Removing the policy barriers to integrated care’, Chris Ham and Judith Smith, 2010. I would also like to see slightly more emphasis on the long-term nature of the efforts to collaborate and the changes experienced over time. The authors explain how the pandemic affected data collection, however I would like more analysis of how the changes wrought by the pandemic to the funding mechanisms affected collaboration – suspending PbR has the potential to have a big impact on how NHS organisations relate to each other, and indicates a big change in the nature of commissioning. Was there any data on this?
--

	Was there data on the various purposes of collaboration? The emphasis in this paper was on control totals for the NHS spend – what other goals (if any) for collaboration came up in the case studies? How did the need to manage control total get discussed in relation to any need to increase those totals/resources available for the ICSs? At the start of the discussion section, the authors state that the shift to collaborative working has been largely welcomed, can they be more precise on who has welcomed this shift (and who has been less welcoming – or perhaps just more sceptical as the authors note in relation to local authorities) Largely welcomed by whom? The suggestion of an independent arbiter is interesting – but to what extent are NHSEI considered independent?
--	---

REVIEWER	Waring, Justin University of Birmingham
REVIEW RETURNED	02-Nov-2022

GENERAL COMMENTS	The paper focuses primarily on the themes of ICS governance and collaboration, specifically how collaborative decision making it achieved given new statutory powers of ICSs The theoretical framing of the paper draws on Ostrom's work on cooperative decision making. The justification for this, given other theories of collaborative governance, could be justified a little more and more specific detail on Ostrom's work especially on the governance of common resources could be further described by setting out relevant concepts and arguments. The focus on permissive policy and sovereignty especially the complication between system and organisational governance are very important points and could be developed further. I would also suggest moving the theoretical paragraph on Ostrom to after these two points to inform the research design and questions. In terms of recent literature on the broad topics it may be worth taking a look at the recent essay by Jones et al in BMJ Leader on collaborative governance and ICS leadership. Although in a parallel field the literature in major system change is also relevant such as Turner et al JHSRP and Fulop et al Imp Sci, and Waring et al Pub Admin Rev Queries When introducing the three layers of ICSs - the middle population level is presented as 'p' - i think using the term 'places' would be better and more consistent stylistically with the presentation of the other levels. Methods I found the initial account of the selection of three cases confusing and would encourage a re-write of this paragraph for the purposes of clarity Query: how did the shift from STPs to ICSs and the pandemic impact data collection?
---

	I would recommend a more developed account of data analysis and how this related to the themes identified in the literature Overall the study approach and design look fine. Results In the findings it is not always clear whether the account given relates to all cases? The text does indicate some variation how and why they differ but there is little in the way of a demarcated case description. I appreciate it is difficult to pull together such rich data in a paper presenting thematic data but i wonder if the findings section could set out the particular issue, them describe how each case addressed it noting common and divergent approaches? I also feel the results could develop some more of the detail, e.g. what types of agreements were difficult, was there unevenness in the localisation of agenda, e.g. were some issues or localities more or less difficult? I assume the authors have far too much data and insight and are trying to cover as much as possible, but is it possible to fine-tune the analysis? And how did the shift to ICSs and the statutory basis of this represent a game-changer? Query: what were the 'softer' mechanisms of holding to account through trust, rather than in a formal or codified way' and how did they evolve? Some of the concluding statements might be over-stating what the study can show? Whilst there was support for collaboration in the three cases it is difficult to suggest that this is the case for the whole NHS, there could be sampling bias? The discussion does a very good of relating the findings to policy change which is very topical and important. However the discussion could engage more with relevant theory whether Ostrom or the likes of Ansell and Gash to explain the data. I was also wondering if the discussion could explain the differences between cases?
--	--

VERSION 1 – AUTHOR RESPONSE

Peer reviewer 1	
The main point that needs to be addressed was the noticeable absence of patients and the public in this paper - where are their voices in ICSs? This needs to be considered, even if simply as a limitation of this paper (in terms data/lack of data in the cases and in analysis as well as PPI in the paper itself).	The study findings in relation to the involvement of patients and the public in ICSs has been added to the 'Reconciliation of system and individual responsibilities' section of the findings (p17), and included in the discussion (p23).

The policy background is very clearly set out, however I would like a little more history included (or relevant citations) about the change from competition to collaboration – are ICS really the marker towards collaboration the authors suggest? I think there is a longer history of trying to balance collaboration and competition that dates back at least to the 2012 Health and Social Care Act, see for example the Nuffield Trust briefing ‘Removing the policy barriers to integrated care’, Chris Ham and Judith Smith, 2010. I would also like to see slightly more emphasis on the long-term nature of the efforts to collaborate and the changes experienced over time.	We agree that collaboration is not a new behaviour in the provision of health and care services, and the English NHS specifically. We have added an acknowledgement of this to the introduction. It is also worthy of note that although collaboration is an enduring feature of the planning and provision of services in the English NHS, the introduction of ICSs and the associated changes in the policy, regulatory and now legislative context represent a significant, and unique, pendulum swing away from competition towards collaboration in the institutional context (even if collaboration has always been present as a behaviour). Therefore we have also included this observation in this section (p4).
The authors explain how the pandemic affected data collection, however I would like more analysis of how the changes wrought by the pandemic to the funding mechanisms affected collaboration – suspending PbR has the potentially to have a big impact on how NHS organisations relate to each other, and indicates a big change in the nature of commissioning. Was there any data on this?	We have data on the changing financial mechanisms in place during our fieldwork and their impact on collaboration. We referred to this in passing in the ‘Reconciliation of system and individual responsibilities’ section (p14), and have added some additional points at p18 -19 in the third findings section ‘Decisions regarding resource allocation being made by systems’
Was there data on the various purposes of collaboration? The emphasis in this paper was on control totals for the NHS spend – what other goals (if any) for collaboration came up in the case studies? How did the need to manage control total get discussed in relation to any need to increase those totals/resources available for the ICSs?	The various purposes of collaboration in ICSs are listed in the first paragraph of the introduction. The control totals were most referred to as they were the most clearly articulated target which could be progressed by ICSs in the relatively short period of system working. The extent to which interviewees reported that their systems had made progress towards addressing the various purposes of collaboration is discussed in the final results section ‘Decisions regarding resource allocation being made by systems’. Table 3 also provides detailed examples of the changes which resulted from collaboration in each of our systems. Some interviewees in our case studies pointed out (as noted in the final results section ‘Decisions regarding resource allocation being made by systems’) that they did not feel that it was possible to achieve financial balance. We have added more information to the findings section ‘Reconciliation of system and

	individual responsibilities' (p15) to explain the views we encountered.
At the start of the discussion section, the authors state that the shift to collaborative working has been largely welcomed, can they be more precise on who has welcomed this shift (and who has been less welcoming – or perhaps just more sceptical as the authors note in relation to local authorities) Largely welcomed by whom?	We have added a qualification regarding various parties' attitudes to collaborative working to the start of the discussion, which reflects the findings reported in the findings section (p20)
The suggestion of an independent arbiter is interesting – but to what extent are NHSEI considered independent?	We have amended this statement to reflect the NHSEI would be independent of local system members (p23).
Peer reviewer 2	
The theoretical framing of the paper draws on Ostrom's work on cooperative decision making. The justification for this, given other theories of collaborative governance, could be justified a little more and more specific detail on Ostrom's work especially on the governance of common resources could be further described by setting out relevant concepts and arguments.	We have included an additional justification of our choice of Ostrom's work as the theoretical framing (pp8-9).
The focus on permissive policy and sovereignty especially the complication between system and organisational governance are very important points and could be developed further.	We have included additional text on p8 highlighting the importance of these factors.
I would also suggest moving the theoretical paragraph on Ostrom to after these two points to inform the research design and questions.	We thank the review this helpful suggestion, which we have enacted.
In terms of recent literature on the broad topics it may be worth taking a look at the recent essay by Jones et al in BMJ Leader on collaborative governance and ICS leadership. Although in a parallel field the literature in major system change is also relevant such as Turner et al JHSRP and Fulop et al Imp Sci, and Waring et al Pub Admin Rev	We thank the reviewer for these references, which were all of interest. We found the references to collaborative governance particularly relevant to this article, and have included them in the revised article (p8).
When introducing the three layers of ICSs - the middle population level is presented as 'p' - I think using the term 'places' would be better and more consistent stylistically with the presentation of the other levels.	We have checked that collaboration at this middle scale is referred to as 'place' throughout.
I found the initial account of the selection of three cases confusing and would encourage a re-write of this paragraph for the purposes of clarity	The paragraph describing the process of shortlisting case study sites has been revised for clarity (p10).
How did the shift from STPs to ICSs and the pandemic impact data collection?	The impact of the pandemic on data collection is described in the revised limitations section of the discussion. In terms of the shift from

	STPs to ICSs, one of our case studies (CS1) was an ICS when the study commenced, and the other two became ICSs during the study. However, this change in status did not impact our data collection in any way as system members, leaders, and the ongoing work of the system remained unaltered. This information has been included in the 'methods' section (p12).
I would recommend a more developed account of data analysis and how this related to the themes identified in the literature.	We have included details of the themes drawn from the theoretical framework and the research questions in the section describing the methods (p12).
In the findings, it is not always clear whether the account given relates to all cases? The text does indicate some variation in how and why they differ, but there is little in the way of a demarcated case description. I appreciate it is difficult to pull together such rich data in a paper presenting thematic data, but I wonder if the findings section could set out the particular issue, then describe how each case addressed it, noting common and divergent approaches?	As the reviewer notes, it is challenging to provide a thematically driven analysis which also demarcates case studies within the confines of a relatively brief article. Many of the challenges that systems experienced (such as decision making in the light of organisational sovereignty) are 'hardwired' into this form of collaboration, and therefore were common across our case studies regardless of differences in local context. The main difference between the case studies was the complexity of system configuration in CS3, and the impact that this had on the development of governance arrangements. This is discussed in the first results section 'Development of decision-making arrangements', where we have added additional text to emphasize this difference (p13). We have revised the results section to clarify when we are reporting findings common across the three case studies and when the findings relate to a specific case study.
I also feel the results could develop some more of the detail, e.g. what types of agreements were difficult, was there unevenness in the localisation of agenda, e.g. were some issues or localities more or less difficult? I assume the authors have far too much data and insight and are trying to cover as much as possible, but is it possible to fine-tune the analysis?	The additional detail added in response to the previous comment about case study demarcation has also addressed this comment.
And how did the shift to ICSs and the statutory basis of this represent a game-changer?	The shift to ICS status did not make a great deal of difference to collaboration, as it did not in itself lead to any change in the decision-making arrangements. One exception is the additional 'transformation' funds received in

	CS2 as a relatively early recipient of ICS status, which we have now referred to in the third results section 'Decisions regarding resource allocation being made by systems' (p18). The statutory changes of HCA 2022 did not occur during the field work period, but their possible implications are explored in the discussion. In particular many of our findings, such as the difficulties with consensus decision making in the light of organisational sovereignty, will remain the same under HCA 2022, as we note in the discussion section.
What were the 'softer' mechanisms of holding to account through trust, rather than in a formal or codified way' and how did they evolve?	The softer mechanisms were the development of the information systems necessary to understand performance, quality and finance across the system, and to facilitate open discussion. This is now detailed on p17.
Some of the concluding statements might be over-stating what the study can show? Whilst there was support for collaboration in the three cases it is difficult to suggest that this is the case for the whole NHS, there could be sampling bias?	As noted in the discussion, our study tends to agree with findings of similar research that has preceded it. We also refer to the strength of our conclusions in relation to the small number of case studies in the section of the discussion dealing with limitations.
The discussion does a very good of relating the findings to policy change which is very topical and important. However the discussion could engage more with relevant theory whether Ostrom or the likes of Ansell and Gash to explain the data. I was also wondering if the discussion could explain the differences between cases?	Ostrom can be usefully deployed to understand the findings and the main difference between the case studies (which was the difficulty agreeing the configuration of the 'place' scale in CS3), and we have revised the discussion to include references to Ostrom's principles for successful management of common pool resources.

VERSION 2 – REVIEW

REVIEWER	Hughes, Gemma University of Oxford, Nuffield Department of Primary Care Health Sciences
REVIEW RETURNED	20-Dec-2022

GENERAL COMMENTS	I think the revisions made in response to reviewers' comments strengthen this paper, its an important and very useful paper on this current configuration of the NHS. a couple of typos/errors have slipped in e.g. on page 5 - the sentence on systems needs rewording
--

REVIEWER	Waring, Justin University of Birmingham
-----------------	--

REVIEW RETURNED	10-Jan-2023
-------------

GENERAL COMMENTS	This paper has been clearly and effectively revised. And it should make a good contribution to policy and debate. Some minor suggestions based upon recent additions to the literature. The contrast with other governance modes in terms of architecture works really well. Could this be picked up in the discussion or conclusion to round off the papers focus on design? The background on Ostrom might be developed further as it is the primary analytical device. It might be worth looking at the recent work by Glenn Roberts that applied Ostrom to health. Also Ansell and Gash's work on collaboration has been shown relevant to ICSs via the recent essay by Lorelei Jones. The recent paper by Waring et al in JHSRP on the micro-politics of system integration is relevant to this discussion especially around the theme of negotiation and agreements around governance and decision-making. This might help draw out some of the deeper issues to consider or offer additional explanatory insight?
--

VERSION 2 – AUTHOR RESPONSE

Comment	Response
Peer reviewer 1	
I think the revisions made in response to reviewers' comments strengthen this paper, its an important and very useful paper on this current configuration of the NHS. a couple of typos/errors have slipped in e.g. on page 5 - the sentence on systems needs rewording	The typo on p5 has been corrected, and the paper has been proof read, and further minor amendments made.
Peer reviewer 2	
The contrast with other governance modes in terms of architecture works really well. Could this be picked up in the discussion or conclusion to round off the papers focus on design?	A sentence stating the value of using approaches based in collaborative governance to understand the development of shared decision making in ICSs, and ways this might be strengthened has been added to the conclusion.
The background on Ostrom might be developed further as it is the primary analytical device. It might be worth looking at the recent work by Glenn Roberts that applied Ostrom to health. Also Ansell and Gash's	The background on Ostrom has been revised and expanded on p8-9.

work on collaboration has been shown relevant to ICSs via the recent essay by Lorelei Jones.	
The recent paper by Waring et al in JHSRP on the micro-politics of system integration is relevant to this discussion especially around the theme of negotiation and agreements around governance and decision-making. This might help draw out some of the deeper issues to consider or offer additional explanatory insight?	Thank you for this helpful suggestion. We have incorporated some of the insights from this paper into the discussion section (p22-23)